# Simultaneous polydirectional transport of colloidal bipeds

Mahla Mirzaee-Kakhki[1], Adrian Ernst [1], Daniel de las Heras [2], Maciej Urbaniak [3], Feliks Stobiecki [3], Jendrik Gördes [4], Meike Reginka [4], Arno Ehresmann [4] & Thomas M. Fischer [1✉]

Detailed control over the motion of colloidal particles is relevant in many applications in colloidal science such as lab-on-a-chip devices. Here, we use an external magnetic field to assemble paramagnetic colloidal spheres into colloidal rods of several lengths. The rods reside above a square magnetic pattern and are transported via modulation of the direction of the external magnetic field. The rods behave like bipeds walking above the pattern. Depending on their length, the bipeds perform topologically distinct classes of protected walks. We design parallel polydirectional modulation loops of the external field that command up to six classes of bipeds to walk on distinct predesigned paths. Using such loops, we induce the collision of reactant bipeds, their polymerization addition reaction to larger bipeds, the separation of product bipeds from the educts, the sorting of different product bipeds, and also the parallel writing of a word consisting of several letters. Our ideas and methodology might be transferred to other systems for which topological protection is at work.

[1] Experimentalphysik X, Physikalisches Institut, Universität Bayreuth, D-95440 Bayreuth, Germany. [2] Theoretische Physik II, Physikalisches Institut, Universität Bayreuth, D-95440 Bayreuth, Germany. [3] Institute of Molecular Physics, Polish Academy of Sciences, 60-179 Poznań, Poland. [4] Institute of Physics and Center for Interdisciplinary Nanostructure Science and Technology (CINSaT), Universität Kassel, D-34132 Kassel, Germany. ✉email: Thomas.Fischer@uni-bayreuth.de

A time-dependent energy landscape can transport objects with different properties into different directions. This is the basic idea behind any sorting process, including sieving. On a microscopic level, an optical lattice can be used to sort particles based on the strength of the interaction between the particles and the lattice sites[1]. Ratchets[2] can also be used for the directed transport of distinct microscopic objects[3]. Simultaneous sorting of heterogeneous materials can be achieved with particles driven through periodically modulated energy landscapes[4]. In general, these mechanisms do not allow to precisely control the direction of the transported objects.

The simultaneous transport of colloidal assemblies that can be adapted to requirements such as the presence or absence of colloidal cargo would allow to switch between multiple tasks on lab-on-the-chip devices. To this end, depending on their intrinsic properties, colloidal assemblies need to respond differently to an externally given command. The analog in computer science is known as a polyglot, a program that can be simultaneously compiled and executed in different languages. Typically, the polyglot performs the same task in all valid languages. However, much more powerful polyglots can be coded to perform different and independent tasks for each language[5]. A current of electrons is the only computational element in computer science. The ability to execute commands in parallel gains relevance if several computational elements are used. For example, in biochemistry the nodes of a metabolic network trigger different reactions in parallel. Other areas that would benefit from using parallel commands include the parallel computing with entangled quantum states[6–8], with DNA oligonucleotides[9–14], and with soft matter devices[15] such as membranes[16], reaction-diffusion computers[17], microfluidic computers[18,19] and colloidal computers[20,21].

Here we develop a parallel polydirectional command for the robust transport of colloidal rods above magnetic patterns. The parallel polydirectional command addresses simultaneously and independently the transport of rods of different lengths, and it is hence more efficient than multiplexing, which addresses one command at a time. We provide a fully explained parallel polydirectional command ultimately based on topological protection.

## Results

**Experimental setup**. Paramagnetic colloidal particles (diameter 2.8 μm) immersed in water are placed on top of a two-dimensional magnetic pattern. The pattern is a square lattice of alternating regions with positive and negative magnetization relative to the direction normal to the pattern, see Fig. 1a. A uniform time-dependent external field of constant magnitude is superimposed to the nonuniform time-independent magnetic field generated by the pattern. The external field induces strong dipolar interactions between the colloidal particles which respond by self-assembling into rods of 2–19 particles.

The orientation of the external field changes adiabatically along a closed loop (Fig. 1b). Despite the field returning to its initial direction, single colloidal particles can be topologically transported by one-unit cell after completion of one loop[22,23]. The transport occurs provided that the loop winds around specific orientations of the external field. In particular, around those orientations given by the unit vectors of the square magnetization pattern[23]. Colloidal rods formed by several particles can also be transported. The rod aligns with the external field since dipolar interactions are stronger than the buoyancy. Hence, if the external field is not parallel to the pattern, one end of the rod remains on the ground while the other one is lifted. As a result, the rods walk through the pattern, see Brownian dynamics

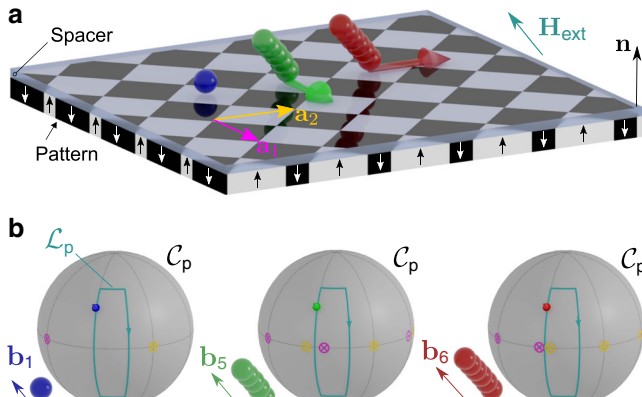

**Fig. 1 Schematic of the colloidal transport. a** A square magnetic pattern with lattice vectors $\mathbf{a}_1$ and $\mathbf{a}_2$ is magnetized with regions of positive (white) and negative (black) magnetization parallel to the vector normal to the pattern $\mathbf{n}$. Spherical colloidal particles (blue) are placed on top of the pattern immersed in water. Due to the presence of a strong homogeneous external field, some colloids self-assemble into rods of different length (green and red). **b** Our control space $\mathcal{C}_p$ is a sphere that represents all possible directions of the external field $\mathbf{H}_{ext}$. The direction of the external field varies in time performing a closed loop $\mathcal{L}_p$. The colored arrow tip on the loop corresponds to the orientation of the external field depicted in **a**. The orientation of the bipeds ($\mathbf{b}_1$, $\mathbf{b}_5$, and $\mathbf{b}_6$) is parallel to $\mathbf{H}_{ext}$. If the loop winds around special directions (yellow and pink equatorial circles), colloidal rods move as bipeds one-unit cell after completion of one loop (**a**). The topological properties of control space depend on the length of the rod, allowing the design of parallel polydirectional loops that move rods of different lengths simultaneously along different directions. Here, the loop transports $b_5$-bipeds with five colloidal particles by $\Delta \mathbf{r}(\mathcal{L}_5) = \mathbf{a}_1$ (green arrow in **a**), $b_6$-bipeds with six colloidal particles by $\Delta \mathbf{r}(\mathcal{L}_6) = \mathbf{a}_2$ (red arrow in **a**), and does not transport single colloids $b_1$.

simulations in Supplementary Movie 1. For this reason, we refer to the rods as bipeds[24–28].

**Parallel polydirectional loops**. To transport the bipeds, the loop also needs to wind around special orientations of the external field. Surprisingly, these special orientations depend on the length of the bipeds, see Fig. 1b. Bipeds of different length fall into different topological classes such that their displacement upon completion of one parallel polydirectional loop can be different in both magnitude and direction. A sketch of the process is shown in Fig. 1. As it is the case for single colloids, the biped motion is topologically protected and hence robust against perturbations. As we show next, it is possible to design parallel polydirectional loops that simultaneously and independently transport bipeds of different lengths.

For example, parallel didirectional loops can be used to simultaneously transport bidisperse bipeds into two different directions. Fig. 2a shows the experimental trajectories of a set of bipeds of lengths $b_5$ and $b_6$, with $b_n = nD$ and $D$ the diameter of the colloidal particles.

Parallel tridirectional loops can be used to initiate the polymerization addition reaction of two bipeds of different lengths by setting them on a collision course and letting the product of the polymerization be transported into a third direction. In Fig. 2b we show experimental trajectories of biped educts of lengths $b_3$ and $b_7$ polymerizing to a product biped of length $b_{10}$.

In Fig. 2c. bipeds of lengths $b_3$, $b_7$, $b_2$, $b_5$, and $b_{10}$ are driven by a complex parallel tetradirectional loop that simultaneously programs the bipeds to write the letters T, E, T, R, and A,

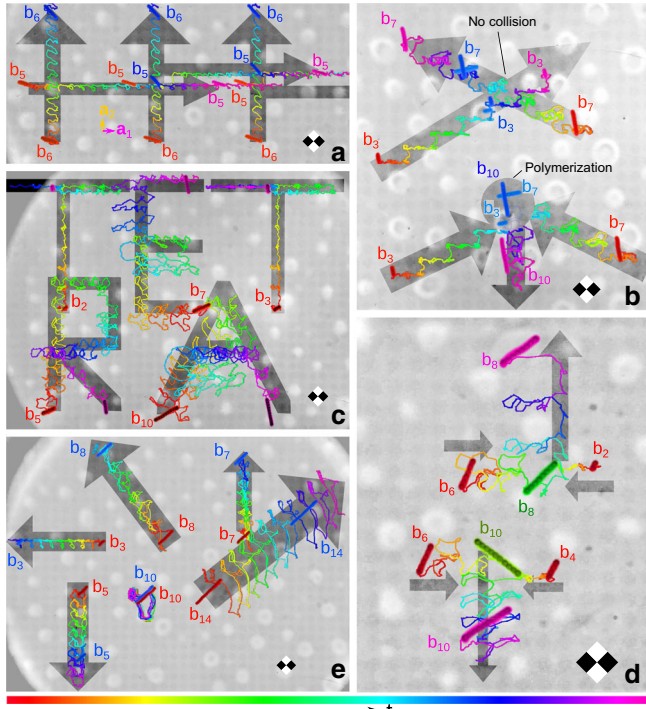

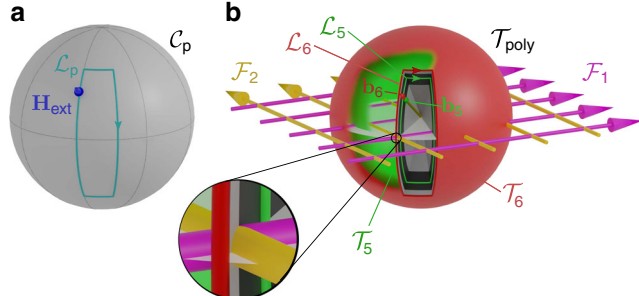

**Fig. 3 Control and transcription spaces. a** Polydirectional control space $\mathcal{C}_{\mathrm{p}}$. Each point on the sphere represents the orientation of the external field. Commands are given as closed loops $\mathcal{L}_{\mathrm{p}}$. **b** Polydirectional transcription space $\mathcal{T}_{\mathrm{poly}}$ contains all unidirectional transcription spaces $\mathcal{T}_n$ that are concentric spheres of radius $b_n$. Each point in $\mathcal{T}_n$ represents the orientation of a biped of length $b_n$. One point in $\mathcal{C}_{\mathrm{p}}$ is transcribed into a ray in $\mathcal{T}_{\mathrm{poly}}$. The unidirectional loops $\mathcal{L}_5$ (green) and $\mathcal{L}_6$ (red) wind around different fence lines and they pass on different sides of the (pink) $\mathcal{F}_1$- and (yellow) $\mathcal{F}_2$-line as can be seen in the magnified inset.

**Fig. 2 Experimental trajectories of bipeds driven by parallel polydirectional loops. a** parallel didirectional loop of robustness $\rho = 0.16$ and compaction $c = 1/2$ transporting several bipeds $b_5$ into the $\mathbf{a}_1$-direction and bipeds $b_6$ into the $\mathbf{a}_2$-direction. **b** parallel tridirectional loop of robustness $\rho = 0.3$ and compaction $c = 7/8$ setting two bipeds $b_3$ and two bipeds $b_7$ on a collision course. One pair of bipeds collides and polymerizes to a longer biped $b_{10}$ that is then transported into a third direction. The non-colliding bipeds continue their motion. **c** Parallel tetradirectional loop of robustness $\rho = 0.2$ and compaction $c = 30/54$ commanding bipeds of five different lengths $\{b_3, b_7, b_2, b_5, b_{10}\}$ to simultaneously write four different letters. The biped $b_2$ is topologically equivalent to the biped $b_3$ and hence writes the same letter (T). **d** Parallel pentadirectional loop of robustness $\rho = 0.11$ and compaction $c = 5/6$ commanding a biped $b_2$ and a biped $b_4$ to polymerize with two colliding bipeds $b_6$ and then separate as they form a biped $b_8$ and a biped $b_{10}$. **e** Parallel hexadirectional loop of robustness $\rho = 0.06$ and compaction $c = 4/8$ transporting six bipeds of different lengths into six different directions. The color of both bipeds and trajectories indicates the time progress of one control loop (see bottom colorbar). A square region of the pattern of diagonal $2a \approx 22\,\mu m$ is sketched in each panel to indicate the scale and the relative orientation.

respectively. Note that bipeds of lengths $b_2$ and $b_3$ perform the same trajectory (letter T). As we show below the reason is that $b_2 < b_3 < a$ with $a \approx 11\,\mu m$ the lattice constant of the pattern. Both $b_2$ and $b_3$ bipeds respond in the same way to the parallel polydirectional loop despite being at different locations. The parallel polydirectional loop does not address the location of the bipeds but only their shape.

We use a parallel pentadirectional loop for the quality control of a competing polymerization addition reaction, Fig. 2d. The loop initiates the addition of a $b_2$ and a $b_6$ biped as well as the addition of a $b_4$ and a $b_6$ biped. Both, the $b_2$ and the $b_4$ bipeds are set on a topologically nonequivalent collision course with $b_6$ bipeds. The products of the addition polymerization reactions are a $b_8$ and a $b_{10}$ biped that are topologically distinct and can be separated from each other as well as from the transport direction of the educts.

In Fig. 2e we plot the trajectories of six bipeds of different lengths after completion of several parallel hexadirectional loops

that transport all six different bipeds consistently into six different directions.

Supplementary Movies 2–6 show walking bipeds with tracked experimental trajectories as well as the driving parallel polydirectional loops. The details of the parallel polydirectional loops are provided in Supplementary Note 1 and Supplementary Figs. 1–3. Supplementary Movie 7 shows the trajectories of the tetradirectional command, panel Fig. 2c, according to Brownian dynamics simulations. The agreement between simulations (Supplementary Movie 7) and experiment (Supplementary Movie 4) is excellent.

**Theory**. As we show next, the reason we can independently and simultaneously transport up to six different particles is topological protection. Let $\mathbf{H}_{\mathrm{ext}}(t)$ be the uniform time, $t$, dependent external field, and $\mathbf{H}_{\mathrm{p}} = \nabla \psi$ the spatially nonuniform and time-independent magnetic field generated by the pattern, with $\psi(\mathbf{r})$ the magnetostatic potential of the pattern, and $\mathbf{r} = (\mathbf{r}_A, z)$ the position vector with components $\mathbf{r}_A$ in the plane parallel to the pattern and $z$ normal to it.

**Control space**. The polydirectional control space $\mathcal{C}_{\mathrm{p}}$ is the surface of a sphere, Fig. 3a, which represent all possible orientations of $\mathbf{H}_{\mathrm{ext}}$. Parallel commands are given in the form of closed loops $\mathcal{L}_{\mathrm{p}}$ in $\mathcal{C}_{\mathrm{p}}$.

**Transcription space**. The orientation of the (dipolar) biped is locked to that of the external field with the northern foot being a magnetic north pole and the southern foot being a south pole. Let $\mathbf{b}_n$ denote the vector from the northern foot to the southern foot of a biped of length $b_n$, Fig. 1a. The southern (northern) foot is on the ground if $\mathbf{H}_{\mathrm{ext}}$ points into the south (north) of $\mathcal{C}_{\mathrm{p}}$. When $\mathbf{H}_{\mathrm{ext}}$ crosses the equator of $\mathcal{C}_{\mathrm{p}}$, the biped is parallel to the pattern and the transfer between the feet occurs. Given the one-to-one correspondence between the biped orientation $\mathbf{b}_n$ and that of the external field, we use $\mathbf{b}_n$ to define unidirectional transcription spaces $\mathcal{T}_n$ given by the surface of a sphere of radius $b_n$. Each point on $\mathcal{T}_n$ corresponds to an orientation of a biped of length $b_n$ and there exists a one-to-one mapping between $\mathcal{C}_{\mathrm{p}}$ and $\mathcal{T}_n$. All unidirectional transcription spaces $\mathcal{T}_n$ can be jointly represented in a polydirectional transcription space $\mathcal{T}_{\mathrm{poly}}$ as concentric spheres of radius $b_n$, Fig. 3b. One point in polydirectional control space $\mathcal{C}_{\mathrm{p}}$ is simultaneously translated into a ray in polydirectional transcription space $\mathcal{T}_{\mathrm{poly}}$. The intersection between the ray and each

unidirectional transcription space is a point that indicates the orientation of the rod. The entire modulation loop $\mathcal{L}_p$ in $\mathcal{C}_p$ is transcribed into a cone with arbitrary cross-section in $\mathcal{T}_{poly}$. The intersection between $\mathcal{T}_n$ and the cone is $\mathcal{L}_n$, the modulation loop for bipeds of length $b_n$.

**Topological classes**. A biped is subject to a total potential dominated by $-\mathbf{H}_{ext} \cdot \mathbf{H}_p$, the coupling between the external and the pattern fields[22,23]. This coupling leads to an effective biped potential $V_n$ proportional to the difference in magnetostatic potential at the two feet. That is,

$$V_n(\mathbf{r}, \mathbf{b}_n) \propto \psi(\mathbf{r} + \mathbf{b}_n/2) - \psi(\mathbf{r} - \mathbf{b}_n/2), \quad (1)$$

with the biped centered at $\mathbf{r}$. Note $V_n$ depends explicitly on $\mathbf{H}_{ext}$ via the one-to-one correspondence between $\mathbf{b}_n$ and $\mathbf{H}_{ext}$. Transport of a biped $b_n$ after completion of one modulation loop $\mathcal{L}_n$ occurs provided that $\mathcal{L}_n$ winds around fences, which are those orientations $\mathbf{b}_n$ for which the potential is marginally stable[23], i.e., the set of biped orientations for which $\nabla V_n = 0$ and $\det(\nabla\nabla V_n) = 0$. For the present square pattern, both conditions are fulfilled along two perpendicular lines in $\mathcal{T}_{poly}$ running through the origin ($b_n = 0$) and parallel to the lattice vector directions. The biped potential is periodic and invariant under the simultaneous transformation $\mathbf{b}_n \rightarrow \mathbf{b}_n + \mathbf{a}_i$ and $\mathbf{r} \rightarrow \mathbf{r} + \mathbf{a}_i/2$ with $\mathbf{a}_i$, $i = \{1, 2\}$, a lattice vector, cf. Eq. (1). The same periodicity in $\mathcal{T}_{poly}$ applies therefore to the fences, which form a square grid of two mutually perpendicular sets of parallel lines separated by one lattice vector, $\mathcal{F}_1$ and $\mathcal{F}_2$, see Fig. 3b. The fences lie in the equatorial plane of $\mathcal{T}_{poly}$ (the plane of biped orientations parallel to the pattern). In each unidirectional transcription space $\mathcal{T}_n$ the fences are points on the equator. Winding around a single fence line of $\mathcal{F}_i$ transports the biped by one lattice vector $\pm\mathbf{a}_i$ depending on whether the loop winds in the same (+) or opposite (−) sense than the axial fence vector. Note that fence lines and lattice vectors are rotated by ninety degrees, cf. Figs. 1 and 2. After completion of a loop $\mathcal{L}_n$, a biped $b_n$ is displaced by

$$\Delta\mathbf{r}(\mathcal{L}_n) = w_n^1(\mathcal{L}_n)\mathbf{a}_1 + w_n^2(\mathcal{L}_n)\mathbf{a}_2, \quad (2)$$

where $\mathbf{w_n} = (\omega_n^1, \omega_n^2)$ is the set of winding numbers of $\mathcal{L}_n$ around the fences $\mathcal{F}_1$ and $\mathcal{F}_2$. Two bipeds $b_n$ and $b_m$ of lengths smaller than the lattice constant $a$ fall in the same topological class since for both of them there exist only two fence lines. Hence, any parallel polydirectional loop $\mathcal{L}_p$ in $\mathcal{C}_p$ is transcribed into modulation loops $\mathcal{L}_n$ and $\mathcal{L}_m$ that have the same set of winding numbers, i.e., $\mathbf{w_n} = \mathbf{w_m}$. However, two bipeds where at least one length is larger than the lattice constant, can be transported independently since it is always possible to find a parallel polydirectional loop in $\mathcal{C}_p$ that is transcribed into two loops $\mathcal{L}_n$ and $\mathcal{L}_m$ with different winding numbers. That is, there exists a parallel polydirectional loop for which $b_n$ and $b_m$ fall into different topological classes. Examples of such loops are shown in Fig. 3.

**Polydirectional degree**. Theoretically and provided that no more than one biped is shorter than the lattice constant, it is always possible to find a parallel polydirectional loop $\mathcal{L}_p$ that transports a collection of bipeds of different lengths independently. We call the number of simultaneously controlled lengths the degree of the parallel polydirectional loop. In Fig. 2 we have shown parallel polydirectional loops of degrees 2–6.

**Robustness**. The limitations in practice arise due to several factors such as the precision of the orientation of the field or deviations due to colloidal polydispersity. Let $\Delta(\mathcal{L}_n)$ be the minimum Euclidian distance from the unidirectional loop $\mathcal{L}_n$ and

all the fences in $\mathcal{T}_{poly}$. Then $\Delta(\mathcal{L}_n)$ provides a direct measurement of the robustness of the transport of a biped $b_n$; the larger value of $\Delta$, the more robust the transport is. We define the robustness of a parallel polydirectional loop of degree $l$ transporting a set of bipeds of lengths $\{b_{n1}, \ldots, b_{nl}\}$ as the minimum value of all individual distances to the fences, i.e.,

$$\rho(\mathcal{L}_p) = \frac{2}{a}\min\{\Delta(\mathcal{L}_{b_{n1}}), ..., \Delta(\mathcal{L}_{b_{nl}})\}, \quad (3)$$

where the prefactor $2/a$ normalizes the robustness such that $0 < \rho < 1$. The robustness decreases with the polydirectional degree, see values in Fig. 2. Experimentally the transport with parallel hexadirectional loops $\mathcal{L}_p$ of robustness as low as $\rho = 0.06$ is still reliable.

**Compaction**. The parallel polydirectional loop is more efficient than multiplexing. That is, addressing sequentially each command, understood as the transport of one biped by one-unit vector. Using parallel polydirectional commands, a single command corresponds to a fundamental loop crossing the equator of $\mathcal{C}_p$ twice. We define the compaction $c$ of a target transport as the ratio of the number of parallel polydirectional commands required and the number of commands in multiplexing. We have implemented a program that optimizes the driving loop by reducing the number of commands, while fulfilling the desired robustness requirements for a given set of target bipeds and displacements. The more robustness we require the less compact the parallel polydirectional loop is. One needs to find a compromise between compaction and robustness. Both the compaction and the robustness are indicated in Fig. 2.

Even though we have not designed tasks that are prompted to be compacted, we achieve a compaction of up to 1/2 and much better values can be obtained for suitable tasks. An example with compaction $c = 1/72$ and robustness $\rho = 0.02$ is shown in Fig. 4 and Supplementary Movie 8, where thirteen bipeds of different lengths between $b_2$ and $b_{19}$ are transported into roughly the same direction but with different magnitudes of the total displacement, which are in all cases commensurate with the lattice constant $a$.

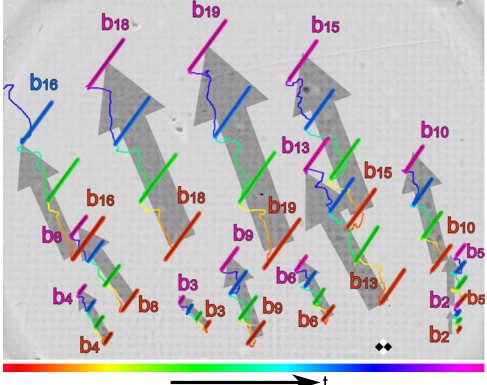

**Fig. 4 Experimental trajectories of bipeds with lengths between $b_2$ and $b_{19}$.** The bipeds are driven by a fundamental loop on a pattern with lattice constant $a \approx 7\,\mu m$ The color of each of the thirteen bipeds and their trajectories indicates the time progress of the repeating control loop (see colorbar). The biped $b_4$ shares the same winding number with the biped $b_6$. The biped $b_{16}$ shares the same winding number with the biped $b_{18}$. The low robustness, $\rho = 0.02$, causes discrepancies between the theoretically computed and the experimentally observed winding numbers for the bipeds $b_{13}$ and $b_{15}$. The compaction of the loop is $c = 1/72$. A square region of the pattern of diagonal $2a$ is sketched for scale and to indicate the relative orientation.

The modulation loop is shown in Supplementary Fig. 4. Although the robustness is low, the experimental and theoretical winding numbers agree with each other. Only for $b_{13}$ and $b_{15}$ the experimental values slightly deviate from the expected winding numbers (error $\Delta w = \pm 1$). Despite this deviation, the modulation can still be used to separate the bipeds but it can no longer be used to steer the bipeds independently into any direction since that requires an exact matching of experimental and theoretical winding numbers.

## Discussion

The complete and simultaneous control over a collection of different objects that we have shown here would be much more difficult to achieve using the ratchet effect. To highlight the differences, consider the transport of stochastic on-off ratchets[2], in which a periodic biased potential consisting of two nonequivalent minima is repeatedly switched on and off. The potential nonergodically confines a particle to a local minimum during the on-phase. When the potential is switched off, the particles can freely diffuse. Effective ratchet-like transport occurs by adjusting the duration of the off-period, since the diffusive time to escape from each potential minimum depends on the escape direction. A ratchet always needs a finite characteristic frequency, determined by the intrinsic dynamics of the system. It is very difficult to adjust several frequencies for different species such that the ratchet effect can work simultaneously for particles of e.g., different shapes. In our approach, which is conceptually closer to a Thouless pump[29] than to the ratchet effect, we also have a potential that is biased by the external magnetic field. However, we avoid ratchet effects and we do not have an off-period. The bipeds always stay in their local minimum to which they are nonergodically confined. Transport occurs solely due to the parametric dependence of the potential on the external field and the enslaved biped direction. Particles follow adiabatically the local minimum at any time during the modulation. Hence, the transport is not the result of a smart choice of dynamic parameters but due to the complexity of the parametric dependence of the topology of the potential on the external field. The transport occurs in the adiabatic limit of vanishing driving frequency. Since the topology is robust, the transport is also robust and feasible in a finite range of frequencies.

Nevertheless, high modulation frequencies must be avoided due to a broadening of the fences. This broadening has almost no effect when manipulating one or two biped lengths, but it makes it very difficult to prescribe the motion of a collection of bipeds due to the occurrence of ratchets. These two-dimensional ratchets lead not only to flux reversal but also to a directional locking into new directions[4]. Another reason to avoid high modulation frequencies is the shape of the rods. The straight shape of the bipeds is due to the dipolar interactions, but the bipeds are quite flexible. High modulation frequencies alter the shape of the rods, changing therefore the topological properties of the transport. This limitation can be alleviated using rigid anisotropic colloidal particles instead of an assembly of colloidal beads.

We have restricted here to rod-like colloidal particles. Elongated enough particles will behave in a similar fashion independently of the precise shape. We expect particles with significant different shapes, e.g., L- or triangular-like colloidal particles, to have completely different topological properties such that it might be possible to control their transport independently.

In summary, using topological protection we have developed a colloidal parallel polydirectional command which delivers an unprecedented control over the simultaneous motion of a collection of colloidal particles. The parallel polydirectional command addresses individually but simultaneously the motion of up to six colloidal bipeds of different lengths. Brownian diffusion does not play any role in our system since the external energy is very large compared to the thermal energy. Hence, it would be possible to construct a macroscopic analog of the system using e.g., an arrangement of NbB-magnets[30]. Downscaling the system from the meso- to the nanoscale is challenging since thermal fluctuations might play a role, broadening the fences and facilitating therefore ratchets.

The parallel polydirectional command paves the way to exciting new applications in colloidal science such as e.g., the automatic quality control of chemical reactions and the parallel computing with colloids. Using directional interparticle interactions, such as in the case of patchy colloids[31], it might be possible to program the assembly of colloids into clusters of complex and predefined shapes that are then transported to the desired direction. Moreover, our methodology and ideas for the simultaneous control of objects that belong to different topological classes can be transferred to other systems for which topological protection is at work.

## Methods

**Pattern**. The pattern is a thin Co/Au layered system with perpendicular magnetic anisotropy lithographically patterned via ion bombardment[32,33]. The pattern consists of a square lattice of magnetized domains with a mesoscopic pattern lattice constant of either $a = 11\,\mu m$ or $a = 7\,\mu m$, see a sketch in Fig. 1a. The magnetic pattern is spin coated with a 1.6 μm polymer film that serves as a spacer.

**External field**. The uniform external magnetic field has a magnitude of $H_{ext} = 4$ kA m$^{-1}$ (smaller than the coercive field of the magnetic pattern) and it is generated by three computer-controlled coils arranged around the sample at 90°.

**Preparation of the initial states**. Paramagnetic colloids are dispersed in water and placed above the magnetic pattern. Without external field the colloids are dispersed mainly as single particles above the pattern. Application of the external field leads to the assembly of random length and random position rods above the pattern. A glass capillary with tip diameter of a few micron attached to a micro manipulator is used to change the length and position of the bipeds to the desired relative initial arrangement while the external field remains in the equatorial plane.

**Visualization**. The colloids and the pattern are visualized using reflection microscopy. The pattern is visible because the ion bombardment changes the reflectivity of illuminated regions as compared to that of the masked regions. A camera records video clips of the bipeds.

**Compacted parallel polydirectional loop**. To find the loop $\mathcal{L}_p$ for given target displacements $\mathbf{w}_{n1}$, …, $\mathbf{w}_{nl}$ for a set of target lengths $b_{n1}$, …, $b_{nl}$, we start by calculating all fence points for each target length. Then, we transcribe the fence points back to control space $\mathcal{C}_p$ by rescaling. All fence points in $\mathcal{C}_p$ lie on the equator (see Fig. 1b yellow and pink circles) and hence they differ only by the azimuthal angle of the external field. For every neighboring fence azimuthal angles we choose one intermediate azimuthal angle between them. These intermediate points are then transcribed to each of the unidirectional transcription spaces of the target set of bipeds and we calculate their minimum distance $\Delta$ to the fences. The intermediate points having transcriptions with $\Delta$ smaller than a fixed threshold are discarded. Every pair formed by two of the non-discarded points defines a fundamental parallel polydirectional loop. Every fundamental parallel polydirectional loop causes a unidirectional displacement for every target length and hence can be represented as a displacement in a $2l$-dimensional vector space of the $l$ target lengths. As the target displacements can be represented in this vector space, the parallel polydirectional loop $\mathcal{L}_p$ can be found as an integer linear combination of fundamental parallel polydirectional loops, i.e., by solving the resulting system of linear equations. Note that only an integer number of every fundamental loop is valid as a solution since only integer winding numbers are possible. If the selected threshold for the robustness is too large, it may be impossible to find an integer solution, but there is always a solution if we sufficiently reduce the minimum robustness. There exist an infinite number of parallel polydirectional loops decoding the same displacements and the one found by solving the linear system of equations is just one of them, not necessarily the most compact. Therefore, we obtain a compaction of this parallel polydirectional loop by searching for pairs (and triplets) of fundamental loops in the parallel polydirectional loop that can be replaced by a single fundamental loop having the same net displacement. This is repeated in an iterative scheme until no further compaction of the loop is found. See Supplementary Movie 9 for a detailed visual explanation of the algorithm.

**Computer simulations**. All parallel polydirectional loops considered here have been also implemented in computer simulations and the colloidal transport matches that found experimentally. Compare, for example, the experimental and simulated trajectories shown in Supplementary Movies 4 and 7, respectively. We simulate the system using overdamped Brownian dynamics. The equation of motion is integrated in time using the standard Euler algorithm. Colloidal particles are modeled as point particles interacting via a Weeks–Chandler–Anderson and a dipolar interparticle potential. Each point particle is subject to the colloidal single particle potential proportional to $-\mathbf{H}_{\text{ext}} \cdot \mathbf{H}_{\text{p}}(\mathbf{r})$.

**Fences**. We express the magnetostatic potential as a Fourier series, see refs. [22,23]. The Fourier modes decay exponentially such that at sufficiently high elevations only the first mode contributes (the spacer forces the colloids to be at high elevations). The fences are found by simultaneously solving the equations $\nabla V_n = 0$ and $\det(\nabla\nabla V_n = 0)$, which for the case of a square pattern is straight forward. See the example for point particles ($b_0$) in ref. [23].

## Data availability

All the data supporting the findings are available from the corresponding author upon reasonable request.

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

## Acknowledgements

This work is funded by the Deutsche Forschungsgemeinschaft (DFG, German Research Foundation) under project number 440764520.

## Author contributions

M.M.K., Ad.E., D.d.l.H., and T.M.F. designed and performed the experiment, and wrote the paper with input from all the other authors. M.U. and F.S. produced the magnetic film. J.G., Ar.E., and M.R. performed the fabrication of the micromagnetic domain patterns within the magnetic thin film.

## Funding

## Competing Interests

The authors declare no competing interests.
