## [Peer Review File · Nature Communications]

REVIEWER COMMENTS

Reviewer #1 (Remarks to the Author):

In this paper, the authors study the response of colloids on a magnetically patterned surface in additional presence of a uniform time-dependent magnetic field. When the external field is varied such, that the intersection point between the magnetic field vector and a surrounding (unit) sphere describes a closed loop, directed and topologically protected transport of particles is achieved. Remarkably, when such loops are chosen appropriately, robust transport of colloidal chains with different lengths into different directions is observed. To prove the unique features of such colloidal transport, the authors demonstrate a model reaction where chains of different lengths merge into larger ones. Depending on the lengths of the product, the resulting chains will be transported into different directions allowing for their separation. The results are in good agreement with computer simulations and are also theoretically explained in terms of general topological arguments.

Topological transport in colloidal systems is a rather novel field with only very few experiments. The manuscript demonstrates this concept to be valid not only for single particles but also for more complex structures. The demonstration of the different behaviors of colloidal rods with different lengths to the same protocol is novel and interesting and may be even useful for future applications. The paper is clearly written and the results are novel and will be of interest to a wide audience. I recommend the paper for publication and have a few minor comments that the authors could address.

1. I have to admit that I found the terminology parallel polyglot (adopted from hyper polyglots) a little exerted, but this may just reflect my personal taste. When subjecting particles with opposite chirality to the same flow field, they are also transported into different directions. Would the authors also refer to that as parallel polyglot behavior? Maybe they consider sharpen their definition.
2. Apart from considering the response of colloidal chains with different lengths, how the trajectories would differ for particles of different shape?
3. I assume, the colloidal chains are rather stiff. How robust is protected transport with similar protocols for semi flexible chains?
4. Inspired by their beautiful example of a model polymerization addition reaction: could their method also used for a hierarchical assembly where single particles assemble into increasingly complex (and topologically different) structures which are then driven together into even more complex structures using the same loop?

Reviewer #2 (Remarks to the Author):

This article propose theoretically and realize experimentally an interesting idea: having several colloidal particles responding differently to the same modulating external field. The interesting part is that this is done in a way that can be scaled to at least 6 different behaviors for 6 different colloidal chains of different length.

I think this work is interesting, timely and deserves publication. The science seems well executed and the idea is original.

I'd like to see the following issues addressed:

- 1) The authors should refer to the literature on Brownian ratchets, where several works have demonstrated sorting and directional control of at least two colloidal species. This literature can be seen as an embryonic form of the idea presented here, although not as sophisticated.
- 2) It'd be good to have the figures with the parallelpolygots employed in fig 2 side by side with the trajectories (maybe split fig 2 into multiple figures?)
- 3) It'd also be good to see simulations videos for the panels in fig 2 similar to suppl video 1.
- 4) I find the title and the analogy with polyglots a bit stretched and missleading. While the science is certainly sound, I'd refrain from such terminology like, polyglot, hyperpolyglot, and colloidal computing. The reference to colloidal computing is particularly out of place and misleading because no computation occurs with the colloids themselves, which are in fact controlled externally.
- 5) Can more than 6 kinds of particles be controlled?
- 6) How does the system control scale with the size of the colloids and the magnetic chessboard? What is the role played by Brownian diffusion?
- 7) I find the technical details lacking in information. The algorithms (for example to find the ideal control cycles) and the theory should be described in much more detail,

Reviewer #4 (Remarks to the Author):

This manuscript describes a quite remarkable approach to controlled transport of colloidal materials through a ratchet-like force landscape. The particles are composed of magnetic spheres bound together into N-particle rods. These particles are allowed to sediment in water onto a checkerboard pattern of oppositely-poled magnetic domains, each domain being larger than a single sphere but smaller than the largest rods. The colloidal rods are then rotated out of the plane and around the normal axis by an external magnetic field whose direction cycles through a fixed pattern. The resulting cyclic force combines with the static force landscape to induce the rods to translate across the substrate. The direction of transport depends on the length of the rod and the symmetry directions selected by the cycling field.

This is an intriguing dynamical system, and the experimental demonstrations of multidirectional transport of mixtures of rods are compelling. The authors invocation of topological production for the selected trajectories similarly is convincing and increases the topical appeal of this work. The paper is well written and beautifully illustrated. I therefore recommend publication in Nature Communications.

I would be satisfied to see the paper published in its present form. Even so, the authors might consider addressing the following three notions.

1. The analogy to polyglots (people and computer programs) is amusing, but can be distracting. The basic idea is that the a time-dependent physical landscape can transport different objects in different directions on the basis of their physical properties. This is how any sorting process works, including sieving. Multidimensional sieves based on optical force landscapes were demonstrated in McDonald, Spalding and Dholakia, Nature 426, 421 (2003). Structured transport and multicomponent sorting by time-dependent ratchets have been widely discussed, including the experimental demonstration by

Skaug, et al., *Science* 359, 1505 (2018). It would be at least as helpful to have the present work placed in the context of previous studies on transport through structured force landscapes as it is to consider the whimsical analogy to polyglots, which ultimately is not particularly enlightening.

2. The graphical construction of trajectories is related to the problem of Buffon's needle. Similar symmetry-based analysis has been used to predict the transport of spheres through periodic potential energy landscapes, for instance in Gopinathan and Grier, *Phys. Rev. Lett.* 92, 130602 (2004).

3. It would be helpful to discuss limitations of the technique. Increasing the cycle frequency, for example, should lead to flux reversals, as it does in many ratchet-like systems. Rotational and translational diffusion might tend to blur out the system's selective transport, but also could introduce additional opportunities, as it does for spheres.

We thank the Reviewers for their time reviewing our manuscript and their useful suggestions that have improved the manuscript. In this point-by-point response letter, we reproduce the comments in *italic* font, give our response in **green-roman font**, and reproduce the changes made to the manuscript using **blue-roman font**. Changes made to the manuscript are also highlighted in **blue** in the new version of the manuscript attached to this resubmission.

Reviewer 1 (Remarks to the Author):

We thank the Referee for the constructive criticism and for the for the positive judgement. We respond to all points raised by the Referee in the following.

In this paper, the authors study the response of colloids on a magnetically patterned surface in additional presence of a uniform time-dependent magnetic field. When the external field is varied such, that the intersection point between the magnetic field vector and a surrounding (unit) sphere describes a closed loop, directed and topologically protected transport of particles is achieved. Remarkably, when such loops are chosen appropriately, robust transport of colloidal chains with different lengths into different directions is observed. To prove the unique features of such colloidal transport, the authors demonstrate a model reaction where chains of different lengths merge into larger ones. Depending on the lengths of the product, the resulting chains will be transported into different directions allowing for their separation. The results are in good agreement with computer simulations and are also theoretically explained in terms of general topological arguments.

Topological transport in colloidal systems is a rather novel field with only very few experiments. The manuscript demonstrates this concept to be valid not only for single particles but also for more complex structures. The demonstration of the different behaviors of colloidal rods with different lengths to the same protocol is novel and interesting and may be even useful for future applications. The paper is clearly written and the results are novel and will be of interest to a wide audience. I recommend the paper for publication and have a few minor comments that the authors could address.

1. I have to admit that I found the terminology parallel polyglot (adopted from hyper polyglots) a little exerted, but this may just reflect my personal taste. When subjecting particles with opposite chirality to the same flow field, they are also transported into different directions. Would the authors also refer to that as parallel polyglot behavior? Maybe they consider sharpen their definition.

No, in that particular case we would not call the transport of particles of different chirality into different directions a "diglot" since the direction cannot be easily controlled. Only the fact that different species are transported into different directions is controlled. Here, we have full control over the direction of the particles, we decide at each time in which direction each species is transported. Nevertheless, we understand the terminology

might not be the most appropriated and in the new version of the manuscript we have sharpened it and replace "polyglot" by polydirectional". Consequently, we have changed "monolingual" by "unidirectional".

Changes made to the manuscript:

Replace "polyglot" by "polydirectional" and "monolingual" by "unidirectional".

2. Apart from considering the response of colloidal chains with different lengths, how the trajectories would differ for particles of different shape?

When colloidal particles self assemble, we sometimes find two bipeds bound to each other side by side such that the total length of the object is shorter than the sum of the individual biped lengths. Those "double bipeds" behave like normal bipeds with the same effective length. J. Martin has shown that bipeds assume an S-shape when subject to a fast rotating external field. We find such S-shaped bipeds when the loop is driven in a non-adiabatic way. Then the S-shaped biped behaves different, but not because of its shape but rather due to ratchet effects that set in due to a broadening of the fences. We are not able to produce other complex shapes when using the dipolar interaction for the self assembly. For example for a triangular shape we would expect the triangle potential to have a form that is a sum of magnetostatic potentials evaluated at the edges of the triangle. Clearly not all three triangle vectors can be enslaved by the external field direction such that figuring out the topology of the potential of such shapes to avoid ratchet effects is much more difficult and we cannot tell exactly if adiabatic trajectories are still possible nor can we tell the protected transport direction by an a priori computation.

We have added the following paragraph to the discussion:

We have restricted here to rod-like colloidal particles. Elongated enough particles will behave in a similar fashion independently of the precise shape. We expect particles with significant different shapes, e.g. L- or triangular-like colloidal particles, to have completely different topological properties such that it might be possible to control their transport independently.

3. I assume, the colloidal chains are rather stiff. How robust is protected transport with similar protocols for semi flexible chains?

Actually the chains are not stiff but super flexible. For example, a biped falls apart into individual colloids as soon as we turn off the external field (see the process at the end of the new Supplementary Movie 8). The rod-like shape is a result of the anisotropic nature of the dipolar interaction. In some sense, our rods can be thought of as flexible polymers stretched by dipolar stresses to its contour length. The apparent rod stiffness shown in the movies and in the figures is because we modulate the external field slowly

enough such that hydrodynamic forces cannot deform the rod-shape into an S-shape.

We have added the following paragraph to the discussion:

Nevertheless, high modulation frequencies must be avoided due to a broadening of the fences. This broadening has almost no effect when manipulating one or two biped lengths, but it makes it very difficult to prescribe the motion of a collection of bipeds due to the occurrence of ratchets. These two-dimensional ratchets lead not only to flux reversal but also to a directional locking into new directions [4]. Another reason to avoid high modulation frequencies is the shape of the rods. The straight shape of the bipeds is due to the dipolar interactions, but the bipeds are quite flexible. High modulation frequencies alter the shape of the rods, changing therefore the topological properties of the transport. This limitation can be alleviated using rigid anisotropic colloidal particles instead of an assembly of colloidal beads.

4. Inspired by their beautiful example of a model polymerization addition reaction: could their method also be used for a hierarchical assembly where single particles assemble into increasingly complex (and topologically different) structures which are then driven together into even more complex structures using the same loop?

The dipolar interaction limits the variety of complex shapes available by self assembly. However, using more complex interactions, such as e.g. patchy colloids with specific bonds that do not enforce the binding to occur into the direction of the external field, such manipulation would be feasible provided that the topology of the effective potential acting on the complex cluster is understood.

We have added the following sentence to the discussion:

Using directional interparticle interactions, such as in the case of patchy colloids [31], it might be possible to program the assembly of colloids into clusters of complex and predefined shapes that are then transported to the desired direction

New Reference added:

[31] Y. Wang, et.al. Colloids with valence and specific directional bonding. Nature, 491, 51-55 (2012).

Reviewer 2 (Remarks to the Author):

We thank the Referee for providing a constructive and positive report and for a thorough reading of our paper. We respond to each point in the following.

This article propose theoretically and realize experimentally an interesting idea: having several colloidal particles responding differently to the same modulating external field.

The interesting part is that this is done in a way that can be scaled to at least 6 different behaviors for 6 different colloidal chains of different length.

I think this work is interesting, timely and deserves publication. The science seems well executed and the idea is original.

I'd like to see the following issues addressed:

1) The authors should refer to the literature on Brownian ratchets, where several works have demonstrated sorting and directional control of at least two colloidal species. This literature can be seen as an embryonic form of the idea presented here, although not as sophisticated.

We thank the referee for this remark. The relation to the ratchet effect is relevant since our approach is quite different, we avoid the ratchet effect and transport the particles adiabatically such that they are always confined to the same minimum of the potential. We discuss now the ratchet effect in the introduction (new introductory paragraph) and also highlight the differences in the discussion.

Changes made to the manuscript

We refer to the ratchet effect in the introduction where we have added two new references:

[2] Reimann P.; Brownian motors: noisy transport far from equilibrium Phys. Rep. 361, 57-265 (2002).

[3] Skaug, M. J., Schwemmer, C., Fringes, S., Rawlings, C. D., Knoll, A. W.; Nanofluidic rocking Brownian motors Science 359, 1505-1508 (2018).

We discuss the differences between the ratchet effect and our approach in the discussion:

The complete and simultaneous control over a collection of different objects that we have shown here would be much more difficult to achieve using the ratchet effect. To highlight the differences, consider the transport of stochastic on-off ratchets [2], in which a periodic biased potential consisting of two non-equivalent minima is repeatedly switched on and off. The potential nonergodically confines a particle to a local minimum during the on-phase. When the potential is switched off, the particles can freely diffuse. Effective ratchet-like transport occurs by adjusting the duration of the off-period, since the diffusive time to escape from each potential minimum depends on the escape direction. A ratchet always needs a finite characteristic frequency, determined by the intrinsic dynamics of the system. It is very difficult to adjust several frequencies for different species such that the ratchet effect can work simultaneously for particles of e.g. different shapes. In our approach, which is conceptually closer to a Thouless pump [29] than to

the ratchet effect, we also have a potential that is biased by the external magnetic field. However, we avoid ratchet effects and we do not have an off-period. The bipeds always stay in their local minimum to which they are nonergodically confined. Transport occurs solely due to the parametric dependence of the potential on the external field and the enslaved biped direction. Particles follow adiabatically the local minimum at any time during the modulation. Hence, the transport is not the result of a smart choice of dynamic parameters but due to the complexity of the parametric dependence of the topology of the potential on the external field. The transport occurs in the adiabatic limit of vanishing driving frequency. Since the topology is robust, the transport is also robust and feasible in a finite range of frequencies.

- New reference [29]

[29] Thouless, D. J. Quantization of particle transport Phys. Rev. B 27, 6083–6087 (1983).

2) *It'd be good to have the figures with the parallelograms employed in fig 2 side by side with the trajectories (maybe split fig 2 into multiple figures?)*

We understand there are some advantages presenting the modulation loops side by side with the trajectories but we also find disadvantages. In particular, we are worried that the complexity of the figure might distract the reader. We provide all modulation loops in the figures of the Supplementary Information. In addition, Supplementary Movies 2-6 and 8 show both the experimental trajectories as well as the modulation loops side by side.

3) *It'd also be good to see simulation videos for the panels in fig 2 similar to suppl video 1.*

We thank the referee for the suggestion. We agree and have made a new video for the most complex case, the tetradirectional loop. The video highlights the excellent agreement between simulations and experiments.

Changes made to the manuscript:

- Added new Supplementary Movie 7 (simulation of a tetradirectional loop).

- We refer to the new Supplementary Movie 7 in the main text and also in the section methods.

4) *I find the title and the analogy with polyglots a bit stretched and misleading. While the science is certainly sound, I'd refrain from such terminology like, polyglot, hyperpolyglot, and colloidal computing. The reference to colloidal computing is particularly out of place*

and misleading because no computation occurs with the colloids themselves, which are in fact controlled externally.

Any computer is externally controlled via software. The polymer addition reaction we demonstrate here is certainly the emulation of an **if - then - else** command that transports the polymer addition product into a third direction if the reaction has been successful and it transports the non reacting educts into the original directions otherwise (else). In any case, we have changed the the title and the polyglot terminology in the new version of the manuscript.

Changes made to the manuscript

New title: Simultaneous polydirectional transport of colloidal bipeds.
Changed “polyglot” terminology to “polydirectional”.

5) Can more than 6 kinds of particles be controlled?

From a strictly theoretical point of view there is no upper limit in the number of particles that can be transported simultaneously provided that the lengths are different. In our experiments, the limitations arise due to e.g. fine details of the pattern and the precision controlling the direction of the external field, which in practice limit the maximum number of bipeds that can be controlled independently. A full control of more than six particles is challenging with our current setup, but we have been able to transport up to 11 bipeds into roughly the same direction but with different magnitudes of the displacements.

Changes made to the manuscript:

- New Fig. 4 showing the motion of 11 types of bipeds of different lengths
- New Supplementary Fig. 4 showing the modulation loop.
- New paragraph added:

An example with compaction $c = 1/72$ and robustness $\rho = 0.02$ is shown in Fig. 4 where thirteen bipeds of different lengths between b_2 and b_{19} are transported into roughly the same direction but with different magnitudes of the total displacement, which in all cases commensurate with the lattice constant a . The modulation loop is shown in Supplementary Figure S4. Although the robustness is low, the experimental and theoretical winding numbers agree with each other. Only for b_{13} and b_{15} the experimental values slightly deviate from the expected winding numbers (error $\Delta w = \pm 1$). Despite this deviation, the modulation can still be used to separate the bipeds but it can no longer be used to steer the bipeds independently into any direction since that requires an exact matching of experimental and theoretical winding numbers.

- Added Supplementary Movie 8 showing the experimental trajectories.

6) *How does the system control scale with the size of the colloids and the magnetic chessboard? What is the role played by Brownian diffusion?*

There are numerous energy scales that define the behavior of the system. The potential energy of the particles scales with the particle volume, independently of the size of the chessboard (the exponential decrease of the potential energy with the elevation is stronger for smaller lattice constants, but the particles feel a universal potential at lower elevations and one can compensate the effect by scaling down the space between the bipeds and the pattern). The scaling down of the dipolar interaction is proportional to the square of the volume of the particles, which makes it harder to keep the bipeds in shape. Most importantly, the fences broaden due to Brownian diffusion and one needs more robustness to avoid thermal ratchet effects. In our system Brownian diffusion does not play any role since the external energy is very large as compared to the thermal energy.

Changes made to the manuscript:

We discuss the effect of up- and down-scaling the experiment:

Brownian diffusion does not play any role in our system since the external energy is very large compared to the thermal energy. Hence, it would be possible to construct a macroscopic analog of the system using e.g. an arrangement of NbB-magnets [30]. Downscaling the system from the meso- to the nanoscale is challenging since thermal fluctuations might play a role, broadening the fences and facilitating therefore ratchets.

- New reference [30]

[30] A. M. E. B. Rossi et al. Hard topological versus soft geometrical magnetic particle transport. *Soft Matter* **15**, 8543–8551 (2019).

7) *I find the technical details lacking in information. The algorithms (for example to find the ideal control cycles) and the theory should be described in much more detail,*

We think the best way to explain the algorithm is visually. We have added a Supplementary Movie that describes the implementation of the algorithm to find the optimal loops.

Changes to the manuscript:

- New Supplementary Movie 9.

- Reference to the movie in the methods section

Reviewer 4 (Remarks to the Author):

We thank the Referee for the positive assessment and for raising several interesting and constructive points. We have addressed these points in the revised version of the paper. A point by point response is provide below.

This manuscript describes a quite remarkable approach to controlled transport of colloidal materials through a ratchet-like force landscape. The particles are composed of magnetic spheres bound together into N -particle rods. These particles are allowed to sediment in water onto a checkerboard pattern of oppositely-poled magnetic domains, each domain being larger than a single sphere but smaller than the largest rods. The colloidal rods are then rotated out of the plane and around the normal axis by an external magnetic field whose direction cycles through a fixed pattern. The resulting cyclic force combines with the static force landscape to induce the rods to translate across the substrate. The direction of transport depends on the length of the rod and the symmetry directions selected by the cycling field.

This is an intriguing dynamical system, and the experimental demonstrations of multidirectional transport of mixtures of rods are compelling. The authors invocation of topological production for the selected trajectories similarly is convincing and increases the topical appeal of this work. The paper is well written and beautifully illustrated. I therefore recommend publication in Nature Communications.

I would be satisfied to see the paper published in its present form. Even so, the authors might consider addressing the following three notions.

- 1. The analogy to polyglots (people and computer programs) is amusing, but can be distracting. The basic idea is that the a time-dependent physical landscape can transport different objects in different directions on the basis of their physical properties. This is how any sorting process works, including sieving. Multidimensional sieves based on optical force landscapes were demonstrated in McDonald, Spalding and Dholakia, Nature 426, 421 (2003). Structured transport and multicomponent sorting by time-dependent ratchets have been widely discussed, including the experimental demonstration by Skaug, et al., Science 359, 1505 (2018). It would be at least as helpful to have the present work placed in the context of previous studies on transport through structured force landscapes as it is to consider the whimsical analogy to polyglots, which ultimately is not particularly enlightening.*

The referral to ratchet landscapes is useful and it helps us explain the different approach that we have followed here. Please see also our response to reviewer 2, point 1. We also would like to emphasize that our work does not simply show the ability of sorting or separating, but the possibility to program the simultaneous transport of the bipeds into any desired direction. It is not applying a field that transport the particles into different directions, but being able to direct the particles into directions provided by someone else a priori without performing an experiment that figures out how to do it. This is possible due to the simplicity and robustness of the complex topology of the biped potential.

Changes made to the manuscript

- New introductory paragraph:

A time-dependent energy landscape can transport objects with different properties into different directions. This is the basic idea behind any sorting process, including sieving. On a microscopic level, an optical lattice can be used to sort particles based on the strength of the interaction between the particles and the lattice sites [1]. Ratchets [2] can also be used for the directed transport of distinct microscopic objects [3]. Simultaneous shorting of heterogeneous materials can be achieved with particles driven through periodically modulated energy landscapes [4]. In general, these mechanisms do not allow to precisely control the direction of the transported objects.

The simultaneous transport of colloidal assemblies that can be adapted to requirements such as the presence or absence of colloidal cargo would allow to switch between multiple tasks on lab-on-the-chip devices. To this end, depending on their intrinsic properties, colloidal assemblies need to respond differently to an externally given command.

- Added four references

[1] McDonald, M. P. Spalding G. C. and Dholakia K.; Microfluidic sorting in an optical lattice; Nature 426, 421 (2003).

[2] Reimann P.; Brownian motors: noisy transport far from equilibrium Phys. Rep. 361, 57-265 (2002).

[3] Skaug, M. J., Schwemmer, C., Fringes, S., Rawlings, C. D., Knoll, A. W.; Nanofluidic rocking Brownian motors Science 359, 1505-1508 (2018).

[4] Gopinathan, A. and Grier D. G., Statistically Locked-In Transport through Periodic Potential Landscapes Phys. Rev. Lett. 92, 130602 (2004).

We discuss the differences with the ratchet effect in the discussion. See response to point 1 of Reviewer 2.

2. The graphical construction of trajectories is related to the problem of Buffon's needle. Similar symmetry-based analysis has been used to predict the transport of spheres through periodic potential energy landscapes, for instance in Gopinathan and Grier, Phys. Rev. Lett. 92, 130602 (2004).

The construction of Buffon's needle is for a needle stochastically placed in a checkerboard. Here our biped is placed at only one deterministic conformation above one of the unit cells. In Buffon's needle the probability of the needle crossing a chessboard boundary is computed. Here we try to avoid the ends of all needles to touch a fence in T_p in order to keep them under control. The deterministic nature of our system allows us a complete control over the motion in contrast to other ratchet-like approaches. We agree symmetry plays a role in both approaches, however we exploit the topology of an almost static potential while the essence of all ratchet approaches lies in the proper adjust-

ment of the ratchet frequency. Our approach works in the adiabatic limit of vanishing modulation frequency.

We added a citation to the work of Gopinathan and Grier [4] and refer to it in the introduction and also in the discussion. See our response to previous point and also the response to point 1 of Reviewer 2.

3. It would be helpful to discuss limitations of the technique. Increasing the cycle frequency, for example, should lead to flux reversals, as it does in many ratchet-like systems. Rotational and translational diffusion might tend to blur out the system's selective transport, but also could introduce additional opportunities, as it does for spheres

We must indeed avoid increasing the frequency since this causes a broadening of the fence. This broadening has no significant effect when transporting one or two biped lengths but it makes it much harder to exploit the topology of the potential by indeed creating ratchet moves that in two dimensions not only lead to a flux reversal but to a directional locking into new directions, as is discussed in e.g. Gopinathan and Grier. Translational and rotational Brownian diffusion do not play a role in our system since the external energy is much larger than the thermal energy. Another limitation is the number of particles of different lengths that can be transported simultaneously.

Changes made to the manuscript.

We describe the limitations caused by high frequency driving in the Discussion (see response to point 3 of Reviewer 1) and the limitations to the number of particles in the Results section (see response to point 5 of Reviewer 2).

Further changes:

- Corrected minor typos.
- Added an abstract as required in Nature Communications.
- Minor stylistic modifications to adhere to the Nature Communications style.

REVIEWERS' COMMENTS:

Reviewer #1 (Remarks to the Author):

The authors have fully addressed my questions and I am happy to support publication of their manuscript.

Reviewer #2 (Remarks to the Author):

The authors have successfully addressed the raised issues. I believe the article can now be published.

Reviewer #4 (Remarks to the Author):

The authors have addressed the reviewers' recommendations from the first round of review. I consider the revised manuscript suitable for publication in Nature Communications in its present form.

Reviewer 1 (Remarks to the Author):

The authors have fully addressed my questions and I am happy to support publication of their manuscript.

Thanks for your work and for the positive feedback.

Reviewer 2 (Remarks to the Author):

The authors have successfully addressed the raised issues. I believe the article can now be published.

Thanks for your work and for the positive feedback.

Reviewer 4 (Remarks to the Author):

The authors have addressed the reviewers' recommendations from the first round of review. I consider the revised manuscript suitable for publication in Nature Communications in its present form.

Thanks for your work and for the positive feedback.